# OL-SLAM: A Robust and Versatile System of Object Localization and SLAM

**DOI:** 10.3390/s23020801

**Published:** 2023-01-10

**Authors:** Chao Chen, Yukai Ma, Jiajun Lv, Xiangrui Zhao, Laijian Li, Yong Liu, Wang Gao

**Affiliations:** 1Institute of Cyber-Systems and Control, Zhejiang University, Hangzhou 310027, China; 2Science and Technology on Complex System Control and Intelligent Agent Cooperation Laboratory, Beijing 100191, China

**Keywords:** SLAM, multi-sensor fusion, object tracking and localization

## Abstract

This paper proposes a real-time, versatile Simultaneous Localization and Mapping (SLAM) and object localization system, which fuses measurements from LiDAR, camera, Inertial Measurement Unit (IMU), and Global Positioning System (GPS). Our system can locate itself in an unknown environment and build a scene map based on which we can also track and obtain the global location of objects of interest. Precisely, our SLAM subsystem consists of the following four parts: LiDAR-inertial odometry, Visual-inertial odometry, GPS-inertial odometry, and global pose graph optimization. The target-tracking and positioning subsystem is developed based on YOLOv4. Benefiting from the use of GPS sensor in the SLAM system, we can obtain the global positioning information of the target; therefore, it can be highly useful in military operations, rescue and disaster relief, and other scenarios.

## 1. Introduction

For autonomous unmanned systems, SLAM technology can observe the surrounding stationary environment and build a 3D map of the environment through sensors such as cameras and LiDARs installed on the robot [1]. For dynamic scenes, SLAM-based 4D reconstruction technology can reconstruct 4D (3D+time) dynamic scenes with rigid moving objects [2,3,4]. However, the complexity of the actual scene means the SLAM system, with only positioning and mapping functions, is unable to meet the needs of many scenarios such as military operations, emergency rescue, and disaster relief. Besides, autonomous unmanned systems are often required to obtain positioning and environmental maps at the same time. It is necessary to develop an intelligent multifunctional perception system with self-positioning, mapping, target tracking, and positioning functions to search for objects of interest within the field of view and obtain the target location.

In order to improve the state estimation accuracy of SLAM systems, a large number of multi-sensor fusion methods have been used, such as the fusion of camera and IMU [5,6,7,8], LiDAR and IMU [9,10,11], and a combination of all of them [12,13,14]. The sensors used in these methods can be divided into local pose estimation sensors such as camera and IMU and global pose estimation sensors such as GPS and magnetometer. However, they all have their advantages and disadvantages, so the single use of a certain type of sensor limits the SLAM system in practical application [15,16,17]. The short-term results are more credible for local pose estimation sensors, but they have two shortcomings. One is that their pose estimation results do not have a global coordinate system, so the method is not reusable. The second is that when the system runs for a long time, there will be an inevitable cumulative drift. Although the loop closure detection method can correct the accumulated error in the SLAM system, problems such as difficult matching, a large amount of data, and limited application scenarios still exist. The frequency of global pose estimation sensors is not high, so they cannot provide much continuous observation information. Furthermore, their measurement noise is relatively large; therefore, they cannot be directly used for pose estimation. However, they have global observation coordinates and are not affected by time accumulation. Therefore, fusing different sensors is an important method to enhance pose estimation accuracy.

For targeting, YOLOv4 [18] provides a high-speed and accurate target detection network model. The optical flow method and Kalman filtering are often used for target tracking. In terms of application, J.A. [19] proposes an automatic expert system, based on image segmentation procedures, that assists in safe landing through recognition and the relative orientation of the UAV and platform. Dr. Krishna [20] specified detection and tracking algorithms in terms of extracting the features of images and videos for security and scrutiny applications. We tend to fuse other sensors such as LiDAR or Radar to locate the target because monocular cameras lack depth information. Yifang [21] proposes the use of RADAR and Infrared sensor (IR) information for tracking and estimating target state dynamics. To project image-based object detection results and LiDAR-SLAM results onto a 3D probability map, Gong et al. [22] combine visual and range information into a frustum-based probabilistic framework.

For the above reasons, this paper proposes an online positioning, mapping, and target-tracking and location system based on camera, IMU, solid-state LiDAR, and GPS. Specifically, our SLAM system consists of the following four parts: a LiDAR-inertial subsystem (LIS), Visual-inertial subsystem (VIS), GPS-inertial subsystem (GIS), and global pose graph optimization (PGO). The LIS and VIS are tightly coupled, and there exists loose coupling between them. The combination of them can improve the accuracy and robustness of the whole system. Finally, the LIS and VIS results are sent to the PGO system for global pose graph optimization to eliminate accumulated drifts. Due to our distributed structure design of tightly coupled internal and loose coupled external subsystems, our system dramatically improved its robustness even in cases where one of the subsystems fails. In addition, a LiDAR–Camera fusion localization method is proposed based on conventional target detection and tracking. The global position of the target is obtained in real-time based on our SLAM.

To test the effectiveness of our system, we built the necessary sensor equipment and collected many scene-rich datasets, including high-altitude UAV aerial photography datasets, ground vehicle datasets, and ground handheld datasets. Considering that there are relatively few datasets that include the sensors we use, we open source all collected datasets for other researchers to use. Finally, we conduct extensive experiments on our dataset to test our system. Experiments show that our system can perform the expected function well with good accuracy and robustness.

The main contributions of this paper are as follows:We propose a high-precision, high-robust multi-sensor fusion online SLAM system;We propose an online target-tracking and localization system based on SLAM results to meet the needs of various natural complex scenes;We collect relevant datasets using our equipment and make the datasets available for other researchers to use.

## 2. Method

Here, we first introduce the block diagram of our system and then introduce our SLAM subsystem and target-tracking and localization subsystem in detail, respectively. Specifically, we first introduce our three subsystems, namely VIS, LIS, and GIS. Then, we describe how to alleviate cumulative drift using global pose graph optimization. Finally, we introduce the object tracking and localization subsystem and demonstrate how to use SLAM system results to obtain the global position of the object.

### 2.1. The Overview of Our System

An overview of our system is shown in Figure 1, which includes a multi-sensor fusion SLAM system and an object tracking and location subsystem. The SLAM system is divided into the following three parts: data preprocessing, three internal subsystems running in parallel, and the final pose-graph optimization. The data preprocessing step preprocesses the input image, IMU, and LiDAR data, including image feature extraction, IMU pre-integration, and LiDAR plane-feature extraction. Then, it will send the results to the three subsystems, i.e., VIS, LIS, and GIS. There is an interaction between the VIS and LIS subsystems. That is, they both provide each other with the current estimated state, which can improve the accuracy and robustness of the whole system. Specifically, for VIS, we refer to the practice of sliding-window-based nonlinear optimization in VINS-Mono [6]. Since the depth of visual feature points of VIS usually has a large uncertainty, inspired by [12], we register the LiDAR point cloud to the image to assist image depth extraction, which significantly improves the accuracy of VIS for feature point depth estimation. For LIS, the large number of LiDAR point clouds leads to significant challenges in the computing performance, so we refer to the approach of ES-IEKF in FAST-LIO2 [10] and use the fast Kalman filter algorithm to accelerate the calculation. For GIS, we use the IMU data for state propagation and the GPS observation data to correct the IMU results to obtain a high-frequency GPS signal equal to the IMU frequency. Finally, we fuse the results of VIS, LIS, and GIS for pose graph optimization to correct the cumulative drift.

Target tracking and localization rely on local point cloud maps and poses of keyframes provided by the SLAM system. First, we detect targets on the image, and track them between consecutive frames using a Kalman filter. Subsequently, we filter the point clouds that fall in the detection frame based on the external parameters of the camera and LiDAR, using Euclidean clustering to filter the portion of the closest target as the target point cloud. Laser points may not occupy the ground target within the camera’s field of view because the laser point cloud becomes sparse with increasing distance. Therefore, we consider using a local point cloud map instead of a single frame of laser points as our input to compensate for the sparsity of the laser point cloud. Finally, the real-time global position of the tracked target is calculated based on the key poses.

### 2.2. Visual Inertial Subsystem

The pipeline of VIS is similar to VINS-Mono [6], and the system block diagram is shown in Figure 1. For VIS, we define the world coordinate system as {W}, and the state variables of the IMU coordinate system are represented in {W} as
(1)$$x_{i}={\left[\begin{array}{ccccc} R_{b_{i}}^{W} & p_{b_{i}}^{W} & v_{b_{i}}^{W} & b_{\omega_{i}} & b_{a_{i}} \end{array}\right]}^{T},$$
where $R_{b_{i}}^{W}\in SO\left(3\right)$ is the rotation matrix, $p_{b_{i}}^{W}\in R^{3}$ is the position, $v_{b_{i}}^{W}$ is the velocity, $b_{\omega_{i}}$ and $b_{a_{i}}$ are the IMU biases. The IMU motion model is as follows:(2)$$\begin{array}{cc} p_{b_{k+1}}^{W} & =p_{b_{k}}^{W}+v_{b_{k}}^{W}\Delta t_{k}\\ \; & +{\;\int \int \;}_{t\in \left[t_{k},t_{k+1}\right]}\left(R_{t}^{W}\left(a_{t}-b_{a_{t}}-n_{a}\right)-g^{W}\right)dt^{2},\\ v_{b_{k+1}}^{W} & =v_{b_{k}}^{W}+\int_{t\in \left[t_{k},t_{k+1}\right]}\left(R_{t}^{W}\left(a_{t}-b_{a_{t}}-n_{a}\right)-g^{W}\right)dt,\\ q_{b_{k+1}}^{W} & =q_{b_{k}}^{W}⊗\int_{t\in \left[t_{k},t_{k+1}\right]}\frac{1}{2}\Omega \left(\omega_{t}-b_{\omega_{t}}-n_{\omega }\right)q_{t}^{b_{k}}dt, \end{array}$$
where
$$\Omega \left(\omega \right)=\left[\begin{array}{cc} -{\left\lfloor \omega \right\rfloor }_{\times } & \omega \\ -\omega^{T} & 0 \end{array}\right],{\left\lfloor \omega \right\rfloor }_{\times }=\left[\begin{array}{ccc} 0 & -\omega_{z} & \omega_{y}\\ \omega_{z} & 0 & -\omega_{x}\\ -\omega_{y} & \omega_{x} & 0 \end{array}\right],$$
$q_{b_{k}}^{W}$ is the quaternion represent of $R_{t}^{W}$, $a_{t}$ is the acceleration reading from IMU, $n_{a}$ is the acceleration noise of IMU, $\omega_{t}$ is the angular velocity reading from IMU, $n_{\omega }$ is the angular velocity noise of IMU, $\Delta t_{k}$ is the duration between the time interval $\left[t_{k},t_{k+1}\right]$.

For the input image, VIS first detects the Fast-corner [23] and then uses the KLT algorithm [24] for optical flow tracking. Since the inverse depth of visual features optimized by VIS has great uncertainty, we accumulate the LiDAR point clouds of recent frames and then project them onto the image to assist depth estimation. Specifically, as shown in Figure 2, we use the transformation between LiDAR and camera coordinate system $T_{L}^{C}=\left[R_{L}^{C}|p_{L}^{C}\right]$ to project the LiDAR point cloud onto the camera image. Then we find the nearest three projected LiDAR points on the image plane for a visual feature by searching a two-dimensional K-D tree. At last, we use these three points to fit a plane and back-project the visual feature onto this plane as its 3D point, which is shown in Figure 3. It can be seen that for most visual feature points, we can accurately estimate their corresponding 3D points. It is very beneficial in improving the accuracy of the VIS.

In the back-end sliding window optimization, the camera pose and the depth of feature points will be optimized as state variables. Unlike the almost completely independent design of LIS and VIS in [12], we use the current state value $x_{L}$ predicted by LIS as the initial state of VIS. The factor is added to the back-end optimization process of VIS, as shown in Figure 4. It is well known that the robustness and accuracy of LIS are higher than VIS, so this design can improve the performance of VIS, thereby making a more accurate estimate of the initial pose for the next LIS.

### 2.3. LiDAR Inertial Subsystem

Our LIS is modified from [10]. Specifically, as shown in Figure 1, we use Iterate Kalman Filter based on error-state to fuse IMU and LiDAR observations. Benefiting from the improvement of Kalman Gain *K* in [10], this algorithm can run in real-time without causing excessive computational burden with the increase in LiDAR observation points. The iteratively optimized Kalman filter algorithm has been proved by [25] to have the same results as the least squares algorithm using Gauss-Newton, so our LIS also guarantees the accuracy of the algorithm.

When receiving a scan from LiDAR, we first extract the plane feature points and then use the poses obtained from inertial integration to remove the motion distortion of the point cloud. We use the IMU state propagation Equation (Equation 2) to obtain an up-to-date estimate of the current LiDAR pose. However, unlike [10], thanks to the existence of our VIS, we can continue to use IMU data to estimate the current LiDAR pose based on the latest pose estimated by VIS. This method can improve the accuracy of our LIS.

After obtaining the pose estimation of current scan, we need to calculate the distance from the extracted plane feature points to the fitted plane, which is same as in [10]. However, in practical applications, the LIS has a significant drift in height, i.e., the *z* axis. So we add a ground constraint to solve this problem and can flexibly choose whether to use this constraint for different scenes. Our ground detection algorithm is simple but effective. Expressly, we assume that the ground is an almost horizontal plane, which is almost always satisfied indoors and holds outdoors in the vast majority of cases. Then we accumulate the last few frames of LiDAR point clouds in the LiDAR coordinate system {L}. We assume that the distance *h* between LiDAR installation height and the ground is unchanged. For cars and handheld devices, this assumption is generally valid. Then we filter out all point clouds whose height is in [h−δ,h+δ], and ground points are almost included. As shown in Figure 5 shows the ground point cloud detected by our algorithm. It can be seen that our algorithm can detect the ground well.

Then we use the RANSAC algorithm to fit ground plane in these point clouds and get the equation of the ground equation in {L}. We can get
(3)$$\left(x_{d}^{L}-p_{d}^{L}\right)\cdot n_{d}^{L}=0,$$
where $x_{d}^{L}$ is point on ground plane, $p_{d}^{L}$ is point on the plane and $n_{d}^{L}$ is the plane normal vector in the {L}. Since world coordinate system {W} is aligned with gravity $g={\left[0,0,-g\right]}^{T}$, we can easily get the actual ground equation in our world coordinates as
(4)$$\left(x_{t}^{W}-p_{t}^{W}\right)\cdot n_{t}^{W}=0,$$
where $p_{t}^{W}=\left[0,0,-h_{0}\right]$ and $n_{t}^{W}={\left[0,0,1\right]}^{T}$, and $h_{0}$ is the distance from the origin of our {W} coordinate system to the ground.

In the same way, we use the current LiDAR pose $T_{L}^{W}$ to convert this actual ground equation into the current frame LiDAR coordinate system {L}. We can get
(5)$$\left(x_{t}^{L}-p_{t}^{L}\right)\cdot n_{t}^{L}=0.$$

Next, we define the detected plane $\left[n_{d}^{L};p_{d}^{L}\right]$, the real plane $\left[n_{t}^{L};p_{t}^{L}\right]$, and the error between them is added to the optimization of LIS to alleviate the *z* axis drift. First we use a rotation matrix R∈SO(3) to rotate $n_{t}^{L}$ to *x* axis, that is
(6)$${\left[\begin{array}{ccc} 1 & 0 & 0 \end{array}\right]}^{⊤}=Rn_{t}^{L}.$$

Then we use this rotation matrix to rotate $n_{d}^{L}$, that is
(7)$${\left[\begin{array}{ccc} x_{d^{\prime }}^{L} & y_{d^{\prime }}^{L} & z_{d^{\prime }}^{L} \end{array}\right]}^{⊤}=Rn_{d}^{L}.$$

We define the variables to measure the orientation error between the detected ground equation and the actual ground equation as
(8)$$\alpha =\arctan\frac{y_{d^{\prime }}^{L}}{x_{d^{\prime }}^{L}},\;\beta =\arctan\frac{z_{d^{\prime }}^{L}}{\sqrt{{x_{d^{\prime }}^{L}}^{2}+{y_{d^{\prime }}^{L}}^{2}}}.$$

Define the variables to measure the translation error of these two planes as
(9)$$\gamma =p_{t}^{L}\cdot n_{t}^{L}-p_{d}^{L}\cdot n_{d}^{L}.$$

We define $e={\left[\alpha ,\beta ,\gamma \right]}^{T}$ as the final ground residual, which can well constrain the error of our system in the directions of roll, pitch, and *z*. This improvement on the final LIS can be seen in our later experiments.

### 2.4. GPS Inertial Subsystem

We add a GPS-inertial subsystem to better prepare for back-end pose graph optimization. In this subsystem, we use the Error-state Kalman Filter to integrate the state obtained by IMU and the state of GPS observations. This fusion can obtain a high-frequency pose output at IMU frequency. The reason for this design is that the frequency of GPS data is relatively low, and it cannot accurately time matched with the output of the VIS or LIS [9,15] both use the assumption of uniform motion to interpolate GPS data. However, in the case of high-speed and non-uniform motion, this assumption will cause the interpolation results to contain large noise, which reduces the accuracy of back-end pose graph optimization. Specifically, we use Equation (Equation 2) to predict the IMU state and define the error of the state x prediction as
(10)$$\delta x={\left[\begin{array}{ccccc} \delta R_{b}^{W} & \delta p_{b}^{W} & \delta v_{b}^{W} & \delta b_{\omega } & \delta b_{a} \end{array}\right]}^{⊤}.$$

For GPS data, it is defined in WGS84 coordinate system. Same as [15], we first convert the data to ENU coordinate system as our state. Assuming that the position of our GPS sensor in the IMU coordinate system {b} is $p_{GPS}^{b}$, then we can get GPS data to observe the origin of IMU coordinate system $p_{GPS}^{W}$ as
(11)$$p_{GPS}^{W}=p_{b}^{W}+R_{b}^{W}p_{GPS}^{b}.$$

Using Equation (Equation 11) we can get the Jacobian matrix of GPS observations for the error state δx as
(12)$$x_{GPS}={\left[\begin{array}{ccccc} I & 0 & -R_{b}^{W}{\left\lfloor p_{GPS}^{b}\right\rfloor }_{\times } & 0 & 0 \end{array}\right]}^{⊤}.$$

Referring to the error-state Kalman filter equation in [26], we can get the result of the fusion of GPS and IMU, which is more accurate than the interpolation of GPS data using the assumption of uniform motion.

### 2.5. Pose Graph Optimization

After all three subsystems complete their estimation tasks, their results are sent to the final pose graph optimization system for processing. In the pose graph optimization system, we select keyframes for the input of VIS and LIS and use iSAM2 [27] to optimize the pose graph. Precisely, we will count the pose changes between the latest keyframes in the relative pose map of the current input frame. If the rotation or translation part of the pose transformation exceeds the threshold we set, then we will use it as an optimized keyframe. Thanks to our GIS, it uses GPS and IMU to perform Error-state Kalman Filter to get high-frequency GPS observations, which makes a GPS observation constraint almost available for each keyframe. However, the GPS signal usually has a large error when occluded, which reflected in our GIS system is the output covariance $P_{GPS}$ is relatively large. We filter the results of $P_{GPS}<\theta_{P}$ to add to pose graph. As shown in Figure 6, it is the block diagram of pose graph optimization system.

### 2.6. Targets Detection and Tracking

We use YOLOv4 to detect the targets and track them with the optical flow method and Kalman filter. We first use the optical flow method to eliminate the deviation of the pixel coordinates caused by the camera movement of targets and then use the Kalman filter to predict their position.

#### 2.6.1. L-K Optical Flow

First, the L-K optical flow assumes that the gray value of the same point in space is constant across images; then, the weighted least squares method is used to estimate the optical flow field, where the grayscale value of the point a=(x,y) at time *t* is assumed to be I=(x,y,t), and the optical flow constraint equation can be derived based on the following assumptions:(13)$$\begin{array}{c} \nabla I\cdot V_{a}+I_{t}=0, \end{array}$$
where $\nabla I=(I_{x},I_{y})$ denotes the gradient of the image at point *a*; and $V_{a}=\left(u,v\right)$ is the optical flow at point a. Assuming that the optical flow is the same at each point in a local neighborhood centered at point a, search for the displacement that minimizes the matching error in this block, i.e., define Equation (Equation 14) for this neighborhood, and minimize its function value as follows:(14)$$F\left(x,y\right)=\sum_{(x,y)\in \Omega }W^{2}\left(x,y\right)\;\left[\nabla I\cdot V_{a}+I_{t}\right],$$
where Ω denotes the local neighborhood of point a and W(x,y) denotes the weight function. The optimal solution of equation Equation (Equation 14) is obtained as follows:(15)$$\begin{array}{cc} A & =\left[\nabla I\left(x_{1}\right),\nabla I\left(x_{2}\right),\cdots ,\nabla I\left(x_{n}\right)\right],\\ W & =\mathrm{diag}\left[W\left(x_{1}\right),W\left(x_{2}\right),\cdots ,W\left(x_{n}\right)\right],\\ b & =-\left[I_{t}\left(x_{1}\right),I_{t}\left(x_{2}\right),\cdots ,I_{t}\left(x_{n}\right)\right]. \end{array}$$

The final equation can be solved:(16)$$\begin{array}{c} V={\left[A^{T}W^{2}A\right]}^{-1}A^{T}W^{2}b. \end{array}$$

The simple L-K optical flow method cannot manage a situation where the UAV is moving quickly at a high altitude. Moreover, it will generate significant computational errors due to the large motion, which will not only affect the algorithm’s accuracy but also reduce the overall computing speed. In this paper, we employ the pyramid-based L-K optical flow method, whose principle is described as follows: First, the optical flow and affine transformation matrices are calculated for the image of the highest layer. The result of the calculation of the previous layer initializes the calculation of next layer. The optical flow and affine transformation matrices are calculated based on the initialization. This process is repeated until the original image layer is reached. The final result will be computed depending on this coarse-to-fine filtering process.

#### 2.6.2. Kalman Filter Update Objective

In the case of a video that needs to be tracked, the state vector can be expressed as follows:(17)$$X=\left[\begin{array}{cccccccc} c_{x} & c_{y} & w & h & v_{x} & v_{y} & v_{w} & v_{h} \end{array}\;\right],$$
where $c_{x},c_{y}$ are centers, w,h are the width and height of the bounding box, and $v_{x},v_{y},v_{w},v_{h}$ are the speed of their change. Note that $c_{x},c_{y}$ are corrected in Section 2.6.1. We can predict $X_{t}$ based on $X_{t-1}$:(18)$$\begin{array}{cc} x^{\prime }= & Fx+\mu , \end{array}$$(19)$$\begin{array}{cc} P^{\prime }= & {\mathrm{FPF}}^{⊤}+Q, \end{array}$$
where F is called the state transfer matrix, P is the covariance of tracking at the moment t−1, and Q is the noise matrix of the system. The observed and predicted values are considered to apply to the same target if they satisfy the following two conditions:The current detection frame is expanded with enough overlap area with the prediction frame;The Reid score is higher than a specific value.

Finally the tracker is updated with predicted ($x^{\prime }$) and observed values, as follows (y):(20)$$\begin{array}{c} y=z-{\mathrm{Hx}}^{\prime }, \end{array}$$(21)$$\begin{array}{c} S={\mathrm{HP}}^{\prime }H^{⊤}+R, \end{array}$$(22)$$\begin{array}{c} K=P^{\prime }H^{⊤}S^{-1}, \end{array}$$(23)$$\begin{array}{c} x=x^{\prime }+\mathrm{Ky}, \end{array}$$(24)$$\begin{array}{c} P=\left(I-\mathrm{KH}\right)P^{\prime }, \end{array}$$
where the observation matrix $Z=[\begin{array}{cccc} c_{x} & c_{y} & w & h \end{array}]$.

#### 2.6.3. LiDAR Vision Fusion Targeting

To calculate the point cloud falling in the bounding box, we first calculate the pixel coordinates of the laser point in the image as follows:(25)$$\left[\begin{array}{c} c_{i}^{cam}\\ 1 \end{array}\right]=T_{b}^{c}{\left(T_{b}^{w}\right)}^{⊤}\left[\begin{array}{c} c_{i}^{map}\\ 1 \end{array}\right],\left(i=1,2,3,\ldots ,N_{A}\right),$$(26)$$p_{i}^{cam}=\left[\begin{array}{c} u_{i}\\ v_{i}\\ 1 \end{array}\right]=\frac{I^{cam}}{w_{i}^{\prime }}c_{i}^{cam}=\frac{1}{w_{i}^{\prime }}\left[\begin{array}{c} u_{i}^{\prime }\\ v_{i}^{\prime }\\ w_{i}^{\prime } \end{array}\right],\left(i=1,2,3,\ldots ,N_{A}\right),$$
where $c_{i}^{map}={\left[x_{i},y_{i},z_{i}\right]}^{⊤}$ is the point in the local point cloud map and $N_{A}$ is the amount of point clouds, $T_{b}^{w}=\left[R_{b}^{w}|p_{b}^{w}\right]$, $T_{b}^{c}=\left[R_{b}^{c}|p_{b}^{c}\right]$, $p_{i}^{cam}$ is the pixel coordinate of the point cloud, and $I^{cam}\in R{}^{3}\times 3$ is the intrinsic matrix of the camera.

Suppose there are *n* trackers in the current image. We filter the point clouds $C^{cam}=\left\{c_{i}^{cam}|i=1,2,3,\ldots ,N_{A}\right\}$,$C^{map}=\left\{c_{i}^{map}|i=1,2,3,\ldots ,N_{A}\right\}$ in the occupancy detection frame and use Euclidean clustering to classify the points in the map point cloud ${\left(C^{map}\right)}^{\prime }$ into *n* classes. Moreover, the center of the point cloud $O=\{o_{i}|i=1,2,3,\ldots ,n\}$ is the target object’s location. The largest rectangle that can wrap the point cloud represents the outline of the target.

## 3. Physical Experiment Analysis

Considering that there is no open dataset that can satisfy all the sensors used by our algorithm simultaneously, we built our hardware equipment and collected a large number of scene-rich datasets to verify our algorithm. First, we will introduce our hardware equipment and the collected datasets. Then, based on the collected dataset, we experimentally verify the improvements made in our system relative to [6,9]. Specifically, our experiments included the improvement of VIS by registering feature depth with LiDAR and improvements related to leading ground constraints into LIS. Then, we tested the improvement of the positioning and mapping accuracy of the SLAM system by adding a GPS to pose graph optimization on datasets of different scales.

We set up two experiments for the target localization system on the ground and in the air, respectively. The ground experiment mainly verifies the Ray-vision fusion’s relative localization effect without providing the global localization error. After that, we conducted aerial experiments while measuring the global positioning error at different distances based on our SLAM. All experiments were performed on the same system with an Intel® Core™ i7-9700 CPU @ 3.00GHz × 8 and Nvidia GTX 1080ti.

### 3.1. Hardware and Dataset of Our System

The hardware of our system is shown in Figure 7, which includes a global shutter camera, a LiVox AVIA LiDAR (FoV of $70.4^{{}^{\circ}}\times 77.2^{{}^{\circ}}$), a GNSS-INS module, a power supply unit, and an onboard computation platform (equipped with an Intel i5-8400 CPU and 16 GB RAM).

We collected various datasets with rich scenes. Specifically, we used two large-scale datasets collected by drones at an altitude of about 100 m, which we call HZ-odom and HZ-map. In the ground scene, we fixed the device to the electric bicycle and collected several datasets, including two large-scale datasets, which we named ZJG-gym and ZJG-nsh. Three medium-sized datasets, which we name ZJG-lib, YQ-odom, and YQ-map. We also collected two hand-held datasets of small-scale scenarios on the ground and named them CSC-build and CSC-road. The specific information of each dataset is in Table 1, which includes the duration of the dataset, the length of the trajectory, whether they include a return to the origin, and the difficulty level.

### 3.2. Feature Depth Registration of VIS

In this experiment, we focus on the improvement achieved by including the depth registration of visual feature points using LiDAR point cloud in our VIS system. Since our VIS system is adapted from VINS-Mono [6], we mainly compare the results of our VIS system for depth registration of visual feature points with [6]. During the experiment, our LIS subsystem only runs the point cloud preprocessing part. It does not use its running results to add priors to VIS. The block diagram of our VIS in the experiment is shown in Figure 8.

We conducted experiments on YQ-map ground dataset. In this dataset, the vehicle runs at a constant speed most of the time; therefore, the IMU is close to degenerating. Moreover, the vehicle has many 90° turns, which means the visual constraints of the VIO may easily fail, and this feature can lead to scale drift problems. For fairness of the experiment, we turned off the loop closure detection thread of VINS-Mono and only compared the accuracy of the odometry. Furthermore, we set VINS-Mono and our VIS system to be identical in terms of front-end feature extraction, back-end sliding-window keyframes, and optimization time. Since the YQ-map dataset returns to the origin, we use the distance from the endpoint to the start point to judge the accuracy. The result is shown in Figure 9. Due to the depth registration of visual feature points, we can see that our system has better scale consistency and higher accuracy than VINS-Mono. This result is easy to explain: our VIS outperforms VINS-Mono due to the extra scale gained by adding LiDAR point clouds to the depth registration of visual feature points.

In addition, to test the improvement of the absolute accuracy of VIS by performing depth registration, we also conduct experiments on large-scale dataset ZJG-gym and use GPS trajectories as ground truth. The comparison results of our VIS and VINS-Mono are shown in Figure 10. We can see that our VIS and GPS trajectories are in better alignment. We use the root mean square error (RMSE) results to measure the result accuracy. The RMSE of our VIS is 5.03 m, while VINS-Mono is 7.92 m.

### 3.3. Ground Constraint of Our LIS

In this experiment, we focus on the improvement by adding ground constraints to our LIS subsystem. Since our LIS system is adapted from FAST-LIO2 [10], we mainly compare the results of our LIS system with ground constraints and [10]. Specifically, in the experiments, we run our LIS alone without VIS’s prediction.

We conducted our experiments on ZJG-lib dataset, which has many horizontal grounds. Specifically, our LIS system uses the same parameter configuration as FAST-LIO2, including the point cloud downsampling density, the number of iterative Kalman filtering, etc. Then in the experiment, we focus on the *z* axis drift. The result is shown in Figure 11. From the figure, we can see that due to the ground constraints we added, our LIS system has almost no drift in the height direction, while FAST-LIO2 shows a significant drift in height. This result is easy to explain. For ZJG-lib dataset, the LiDAR installation location is close to the ground with a height of 0.8 m. Therefore, the incident angle of the LiDAR scanning distant ground points is small, which reduces the accuracy of the point cloud scanned by LiDAR and increases the drift in altitude of FAST-LIO2.

### 3.4. Pose Graph Optimization of Our System

In this experiment, we focus on the improvement of our entire SLAM system due to the addition of global pose graph optimization. To demonstrate the cumulative trajectory drift suppression that occurs when incorporating a GPS, we conducted experiments on large-scale datasets in the air and on the ground. Specifically, we used multiple large-scale datasets to examine the improvement of our system by fusing GPS data.

First, we conduct experiments on an aerial dataset with HZ-map dataset, in which the drone carries equipment to collect data at an altitude of 100 m. Because of the poor weather conditions when collecting the data, the aircraft in the air is highly unstable, so this dataset presents a considerable limitation in terms of the accuracy and robustness of the SLAM system. Here, we compare the localization and mapping of our system with VINS-Mono and FAST-LIO2 systems, and the results are shown in Figure 12. Since our system fuses GPS data, there is better consistency in positioning and mapping results in large-scale scenes. However, due to the turbulence of the drone at a high altitude in this dataset, VINS-Mono fails and does not provide meaningful results.

In order to better test the accuracy and robustness of our algorithm, we also conducted experiments on many other datasets and used the GPS trajectory as the ground truth. We show the root mean square error (RMSE) results in Table 2 and some trajectory results in Figure 13. The compared algorithms are the separate VIO system VINS-Mono [6], the separate LIO system FAST-LIO2 [10], and the state-of-the-art LIVO systems R2LIVE [13] and R3LIVE [28], which are most similar to our system. It is clear that our system achieves the best accuracy and the most robust performance. Interestingly, after R2LIVE [13] and R3LIVE [28] are integrated with cameras, some data sequences have a lower precision than FAST-LIO2 [10]. This is expected because incorporating visual information in situations that are not conducive to camera work may reduce accuracy. Therefore, visual information is usually incorporated in LIO systems mainly to increase the robustness of the whole system. In addition, R2LIVE [13] and R3LIVE [28] failed on both HZ-map and HZ-Odom datasets, which are run in harsh high-altitude environments. Their two subsystems, VIO and LIO, are tightly coupled. Once a subsystem state is incorrectly estimated, it will have a devastating impact on the entire system. Because our VIO and LIO subsystems are loosely coupled, the collapse of one subsystem will not affect the regular operation of the whole system.

### 3.5. Robustness Evaluation of SLAM System

First, we evaluate the robustness of our system when subject to severe motion. HZ-map is a dataset of drones flying in the air. During automatic flight, the plane experiences sudden stops, turns, and other actions. As shown in Figure 14, the plane has a large number of drastic pitch and yaw angle changes. It can also be seen from the previous experiments that VINS-Mono, R2LIVE, and R3LIVE all failed. Our VIS subsystem also failed. However, the other two subsystems still usually work, which showes the excellent robustness of our system.

Then, we tested the performance of our system when the sensor failed on YQ-map dataset. As shown in Figure 15a, we selected three locations uniformly throughout the trajectory and disabled the camera, LiDAR, and GPS to detect the impact on the system. The experimental results are shown in Figure 15b. When one or two sensors fail, our system can operate normally and obtain relatively accurate trajectory results. Running only the VIO system leads to poor trajectory results only when both the GPS and LiDAR fail. This experiment proves that our system has good robustness.

### 3.6. LiDAR-Vision Fusion Relative Localization

One person was arranged in an open outdoor scene as the target to be located in the experiment. The person holds RTK, as shown in Figure 7, facing a dynamic target walking arbitrarily within 10–40 m of the device’s field of view (as shown in Figure 16). We randomly select some locations as checkpoints and use the laser to measure the distance between the target and the sensor module to compare with the localization results and evaluate the relative localization accuracy of the algorithm. The experimental results are shown in Table 3.

As seen from the results in Table 3, the proposed LiDAR-Camera fusion positioning method in this paper has high measurement accuracy: the relative positioning error is provided in centimeters when the target is less than 40 m away from the measurement unit. As GPS is usually only accurate to the meter level; therefore, it is essential first to ensure that the relative positioning accuracy is as high as possible to maintain the lowest possible error when converting the object’s position to the world coordinate system. The following describes the global positioning experiment for the target.

### 3.7. SLAM Based Global Localization

We chose to perform this experiment in the air, keeping with the system’s actual application scenario. Again, we chose people as targets to evaluate our algorithm. As shown in Figure 17, our UAV flies to a distance of 10 m, 30 m, 60 m, and 90 m from the target for ground reconnaissance, while on the other hand, the target on the ground carries a GNSS receiver that moves at least ten meters along a given trajectory. At this point, the UAV is stationary or follows the target in motion, keeping the target in view. Once the separation distance exceeds 60 m, it is difficult to observe the target in the image with the naked eye. Therefore, we retrained the detection model using the Visdrone Dataset [29] and our small air-to-ground target dataset (Table 4).

On the other hand, the laser point cloud at high altitudes would be sparse in terms of measuring the distance to the target. Therefore we use the local point cloud map provided by our SLAM system for the calculation instead of the single frame. In addition, our SLAM system also provides the UAV pose corresponding to the local point cloud, which can be used to calculate the global positions of the targets. It is worth mentioning that our framework locates all targets in the field of view in real time. However, to facilitate the evaluation of the results, we only count the localization error of one of the targets (Figure 18). The RTK conversion on our target path point to the coordinates under the takeoff point ENU coordinate system is used as ground truth. The distance between the localization result and its closest ground truth is used as the error to evaluate the algorithm.

We collected data using a UAV and verified our algorithm offline. The target position was inferred in 71.5 ms in one round of the experiment.The experimental results are shown in Table 5. Even at high altitudes where the sensors moved violently, our algorithm tracked the target stably and maintained high positioning accuracy. Especially at medium and long distances of 90 m, where the target occupied only a dozen pixel values, our system maintained an error of about one meter.

## 4. Discussion Conclusions

In the case of GPS denial, UAV self-positioning and target detection technology can play a very influential role in the military and rescue fields that require reconstructing the target area’s ground scene quickly, obtaining the corresponding GPS position, and marking the category and real-time position of some critical targets.

This paper proposes a robust, versatile self-localization mapping and target-tracking localization system. Our SLAM system fuses multiple local and global sensors, including camera, LiDAR, IMU, and GPS, and thus has the advantages of high local accuracy and no global drift. Our SLAM system consists of three subsystems, VIS, LIS, and GIS. The three subsystems are tightly coupled, and the subsystems are loosely coupled through information sharing. This system architecture not only ensures the system’s accuracy but also improves the system’s robustness. We built our experimental equipment, collected many air and ground datasets, and conducted detailed experimental verification and analysis. The experimental results prove that our SLAM system has higher accuracy and better robustness than the current SOTA system. We also introduce a LiDAR-Camera Fusion object tracking and localization algorithm. We first used the retrained YOLOv4 to detect the target’s position on the image and used LK optical flow and the Kalman filter to track the targets. We used LiDAR to recover the depth Information of the Target. Our target-tracking and localization system can effectively detect the target of interest and perform global localization of the target based on the results of the SLAM system. We conducted extensive experiments on our collected datasets, and the results show that our system performs the expected functions well. In the future, we will research real-time online tightly coupled GPS data and the high-altitude detection of small targets.

## Figures and Tables

**Figure 1 sensors-23-00801-f001:**
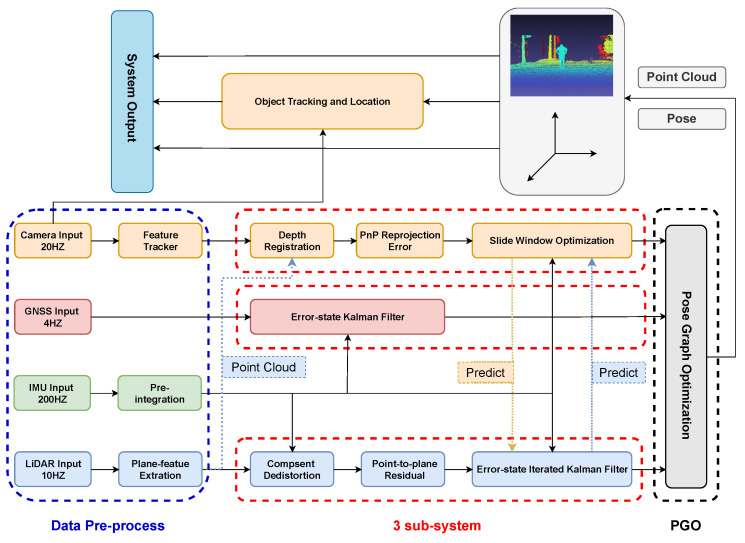
The overview of our system.

**Figure 2 sensors-23-00801-f002:**
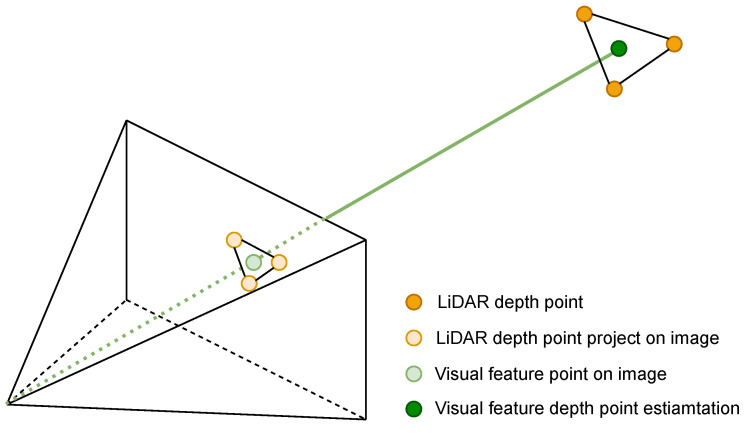
Visual feature depth registratoin with LiDAR points.

**Figure 3 sensors-23-00801-f003:**
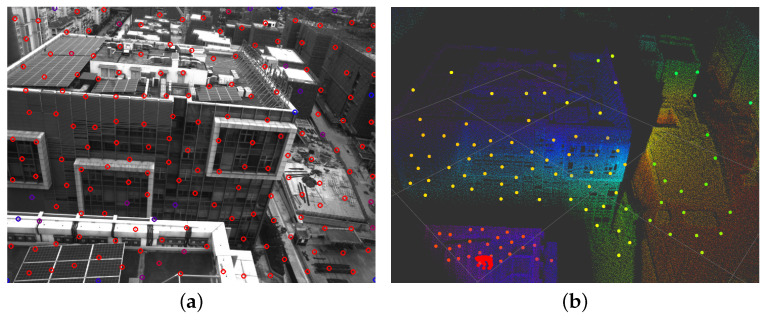
(**a**) The visual features on the image. (**b**) The estimated 3D points of visual feature. It’s clear that lots of visual features can get accurate estimates of their 3D points.

**Figure 4 sensors-23-00801-f004:**
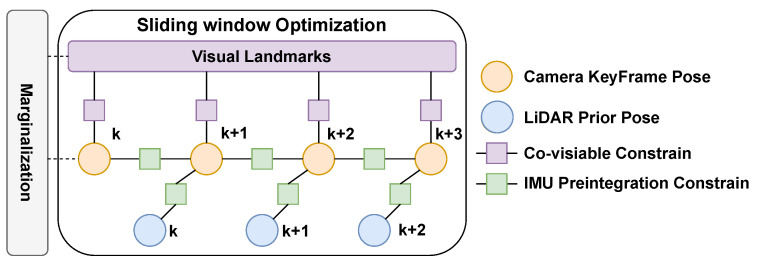
The factor graph of our visual inertial subsystem.

**Figure 5 sensors-23-00801-f005:**
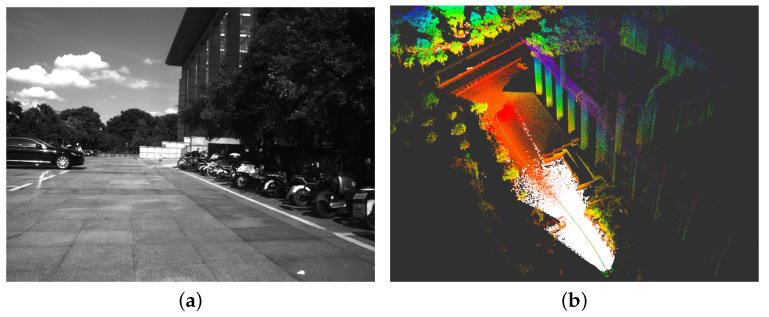
(**a**) The visual image view. (**b**) The ground point clouds(white) detected by our LiDAR inertial subsystem. It’s clear that the ground point clouds can be segmented successfully.

**Figure 6 sensors-23-00801-f006:**
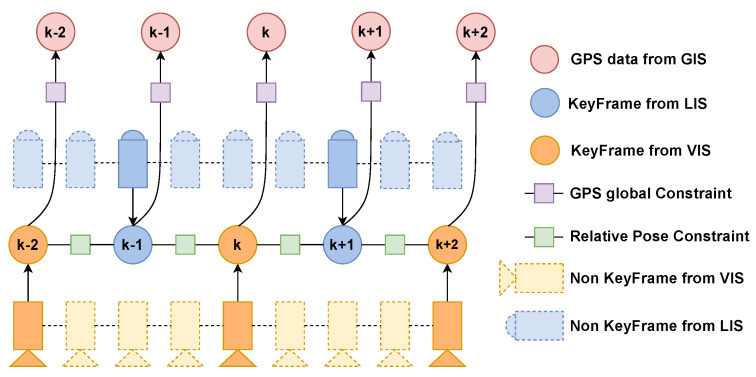
The block diagram of pose graph optimization.

**Figure 7 sensors-23-00801-f007:**
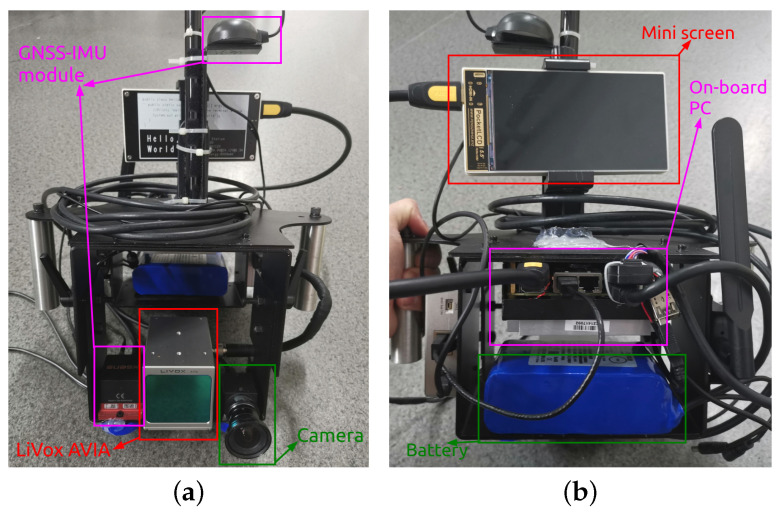
(**a**) The front view of our hardware. (**b**) The back view of our hardware. The total weight of our device is below 3 kg.

**Figure 8 sensors-23-00801-f008:**
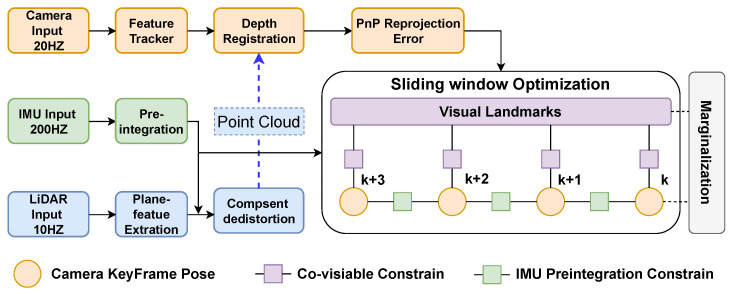
The block diagram of our VIS experiment.

**Figure 9 sensors-23-00801-f009:**
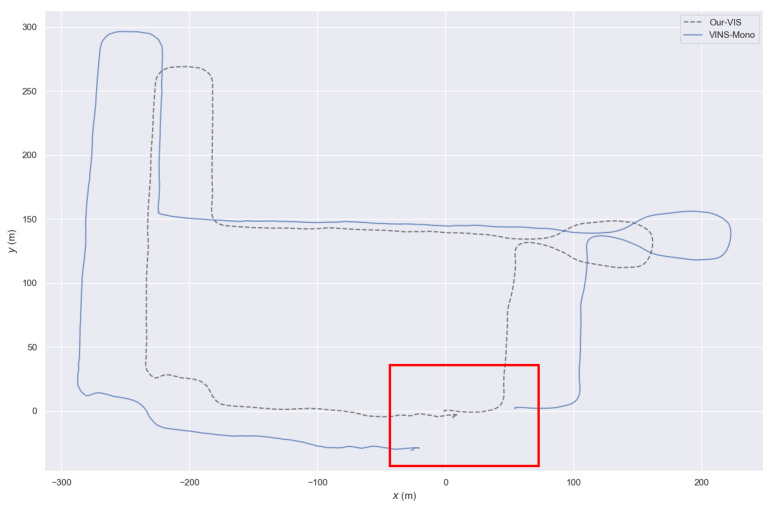
The result of our VIS and VINS-Mono on YQ-map dataset. It goes back to origin.

**Figure 10 sensors-23-00801-f010:**
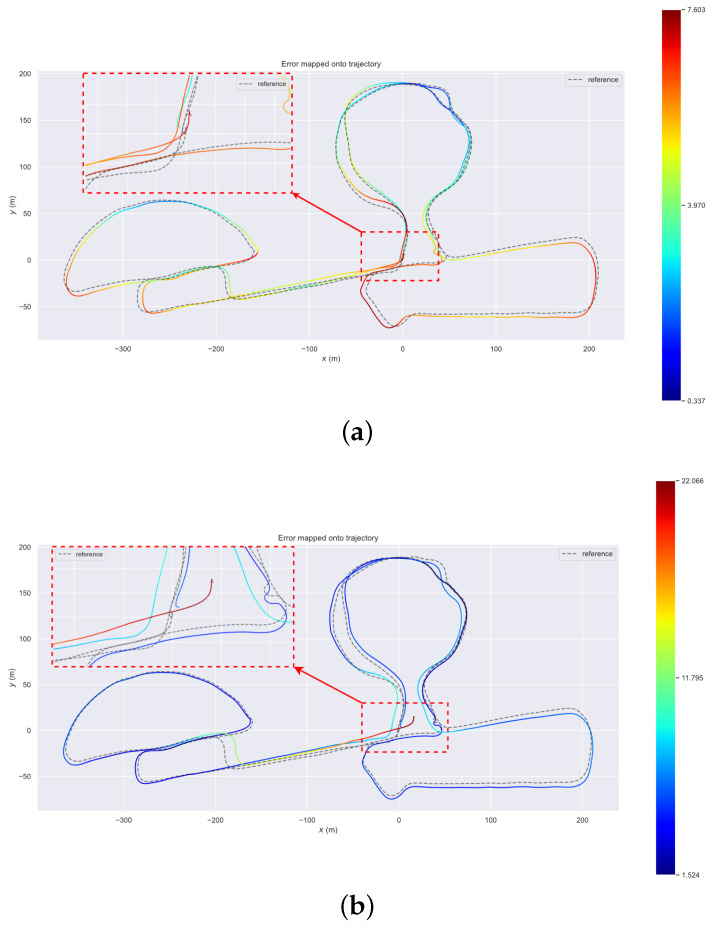
(**a**) The trajectory alignment of our VIS and GPS on ZJG-gym dataset. (**b**) The trajectory alignment of VINS-Mono and GPS on ZJG-gym dataset. It is clear that our VIS has a better result.

**Figure 11 sensors-23-00801-f011:**
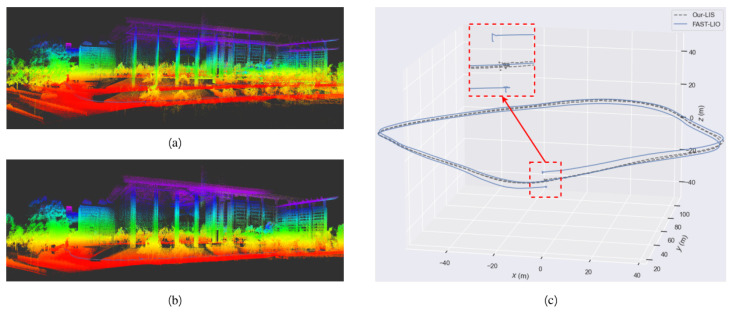
(**a**) The map result of FAST-LIO2 on ZJG-lib dataset. (**b**) The map result of our LIS with ground constraint on ZJG-lib dataset. (**c**) The path result of FAST-LIO2 and our LIS. It is clear that due to the ground constraint, our LIS has less drift than FAST-LIO2.

**Figure 12 sensors-23-00801-f012:**
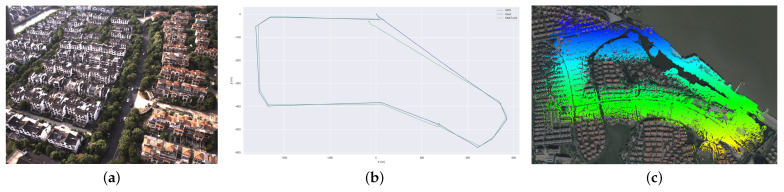
(**a**) The image view of our HZ-map dataset. (**b**) The trajectory of GNSS, FAST-LIO2 and our system. VINS-Mono failed due to the aggressive motion. (**c**) The map built by our system aligned with Google Earth. As a result of the GNSS fusion in our system, we obtain a highly accurate result.

**Figure 13 sensors-23-00801-f013:**
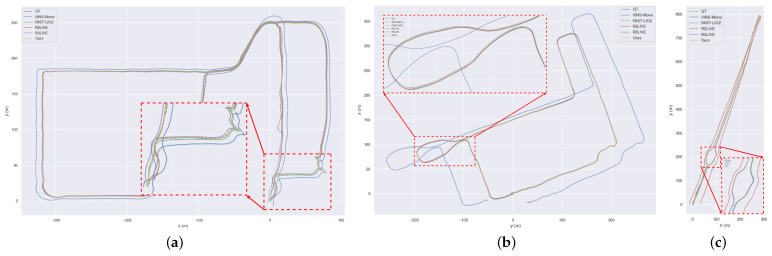
(**a**) The trajectory results on ZJG-nsh dataset. (**b**) The trajectory results on YQ-map dataset. (**c**) The trajectory results on YQ-odom dataset.

**Figure 14 sensors-23-00801-f014:**
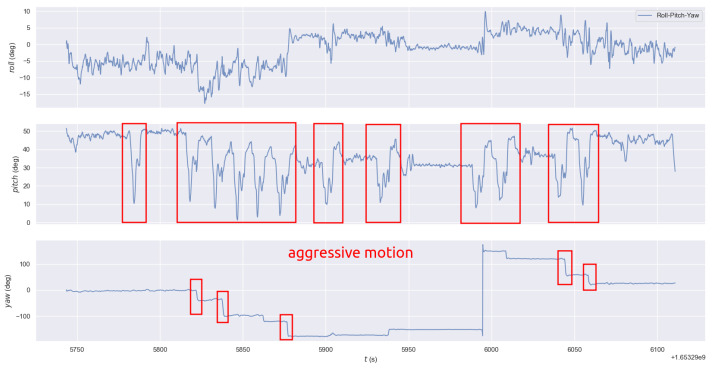
The Euler Angles of HZ-map dataset, which has an aggressive motion.

**Figure 15 sensors-23-00801-f015:**
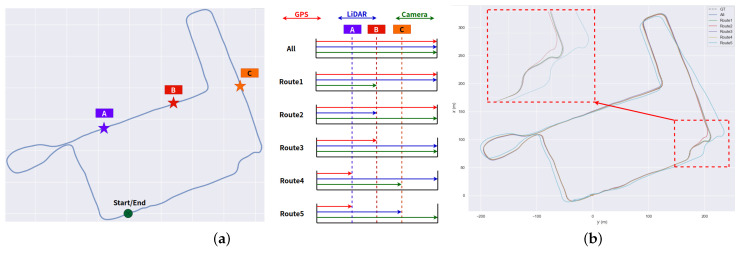
(**a**) Three locations throughout the trajectory to disable the camera, LiDAR, and GPS. (**b**) The trajectory result of our system when one or more sensors are closed.

**Figure 16 sensors-23-00801-f016:**
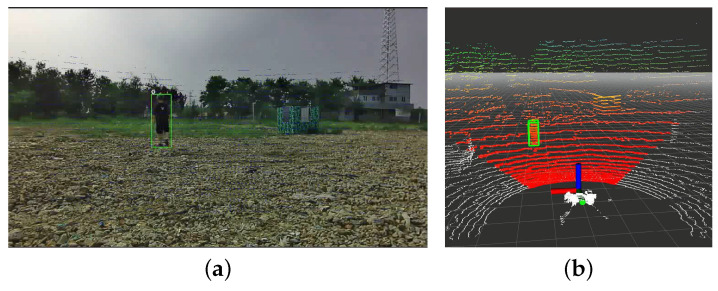
Two test screenshots are shown. (**a**) The results of image detection and tracking. The target is marked with a green box, and the point cloud in (**b**) is also marked with a green box to indicate the spatial position relative to the perception module.

**Figure 17 sensors-23-00801-f017:**
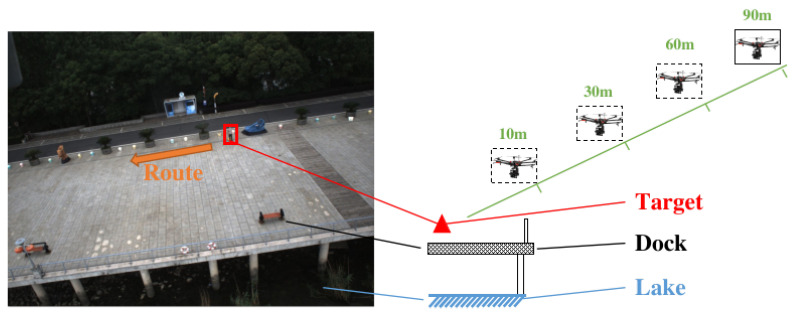
Schematic diagram of the aerial reconnaissance experiment. The image on the left is the actual aerial image (about 30 m apart), the red box is the target’s position, and the image on the right indicates the relative position relationship between the UAV and the target.

**Figure 18 sensors-23-00801-f018:**
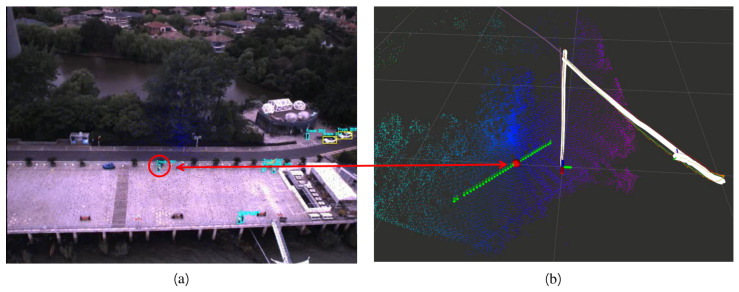
The display image of the tracking effect. (**a**) shows the real-time tracking result, and after a target is selected, the target is indicated by a red ball in (**b**). The green in (**b**) represents ground truth, and the white color is the aircraft’s trajectory calculated by our SLAM.

**Table 1 sensors-23-00801-t001:** Dataset detail.

Dataset	Duration (s)	Length (m)	Loop Closure	Difficulty Level
HZ-odom	280	2380	No	Difficult
HZ-map	367	3068	No	Difficult
ZJG-gym	612	2761	Yes	Difficult
ZJG-nsh	576	1668	Yes	Difficult
ZJG-lib	239	866	Yes	Medium
YQ-odom	306	1718	Yes	Medium
YQ-map	256	1298	Yes	Medium
CSC-road	80	117	Yes	Easy
CSC-build	73	86	Yes	Easy

**Table 2 sensors-23-00801-t002:** RMSE translation error w.r.t GPS.

Algorithm	HZ-Map	HZ-Odom	YQ-Map	YQ-Odom	ZJG-Gym	ZJG-Nsh	ZJG-Lib	CSC-Road	CSC-Build
VINS-Mono [6]	Failed	Failed	50.23	2.34	7.92	6.38	39.00	0.57	0.66
FAST-LIO2 [10]	15.20	0.96	1.80	1.34	3.30	1.88	3.22	0.35	0.43
R2LIVE [13]	Failed	Failed	1.70	12.18	3.36	1.95	3.46	0.26	0.37
R3LIVE [28]	Failed	Failed	1.68	1.37	2.95	2.04	3.78	0.28	0.34
Ours	**0.72**	**0.55**	**0.75**	**0.75**	**2.94**	**0.97**	**1.60**	**0.24**	**0.29**

**Table 3 sensors-23-00801-t003:** Relative Positioning error.

No.	x (m)	y (m)	$\sqrt{x^{2}+y^{2}}$ (m)	Truth (m)	Error (m)
1	15.19	−1.70	15.285	15.30	0.015
2	18.63	−2.43	18.788	19.20	0.412
3	24.76	3.56	25.015	25.0	−0.015
4	25.96	−3.68	26.220	26.30	0.0.8
5	28.82	−9.97	30.496	30.50	0.004
6	35.84	−1.41	35.868	35.90	0.032
7	26.47	4.33	26.821	26.80	−0.021
mean	-	-	-	-	0.073

**Table 4 sensors-23-00801-t004:** Comparison of Visdrone and our Dataset.

Image Object Detection	Scenario	Images	Categories	Avg. Labels/Categories	Resolution (m)	Occlusion Labels
Visdrone [29]	drone	10,209	10	54.2 k	2000 × 1500	✓
Ours	drone	3625	2	13.1 k	1440 × 1080	✓

**Table 5 sensors-23-00801-t005:** Global Positioning error.

Distance (m)	Amount	Min (m)	Max (m)	Median (m)	SD (m)	MAE (m)
10 (±1)	268	0.16	1.86	0.63	0.30	0.70
30 (±1)	200	0.26	3.63	0.99	0.54	1.08
60 (±2)	216	0.07	2.19	0.71	0.35	0.74
90 (±3)	158	0.02	3.91	1.09	0.59	1.11

## Data Availability

Not applicable.

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
