# Peer review of "OL-SLAM: A Robust and Versatile System of Object Localization and SLAM"

_sensors, 2023, doi:10.3390/s23020801_

Round 1
Reviewer 1 Report
The authors have proposed a SLAM and object tracking algorithm with multiple sensors fusion. Although the manuscript contains somewhat novelity but the provision of datasets makes it more appealing. I have following observation before the manuscript is accepted for publication,
1. The manuscript title needs to be modified localization instead of location will be a more appropriate word. The title overall can be modified and will improve readership of the paper.
2. I would suggest the authors to provide more literature on the subject. Specifically the literature survey on tracking is not sufficient.
3. Although the manuscript uses real data from sensors, a comparison with any existing technique or more detail regarding the experimental setup and execution of the algorithm is necessary. For instance is the algorithm real time? what platform has been used? How were the authors able to make the yolo algorithm run in real time and what was the frame rate or the execution time? training data etc must be presented in tabular form regarding the specifics mentioned in the para above.
4. the conclusion may be expanded and should provide more detail.
I would suggest the authors to look into scenarios where the gps signal is denied, how does the system cope with such scenarios? Cite some relevant papers like,
https://doi.org/10.3390/s19245357
https://doi.org/10.1016/j.inffus.2020.10.018
tracking:
https://doi.org/10.3390/s20143821
https://doi.org/10.1016/j.eswa.2017.01.017
https://doi.org/10.1117/1.OE.54.5.053110
10.1109/TAES.2008.4655350
Yolo in tracking
10.1109/ICIRCA51532.2021.9544598
https://papers.ssrn.com/sol3/papers.cfm?abstract_id=4274384
Author Response
Thank you for reading, please open the attachment for details!

Reviewer 2 Report
Dear Authors,
Find attached file for reviews.
Regards;

Author Response
Thank you for reading, please open the attachment for details
